

# Drought stress detection technique for wheat crop using machine learning

Ankita Gupta[1], Lakhwinder Kaur[1] and Gurmeet Kaur[2]

[1] Computer Science and Engineering, Punjabi University, Patiala, Punjab, India
[2] Electronics and Communication Engineering, Punjabi University, Patiala, Punjab, India

## ABSTRACT

The workflow of this research is based on numerous hypotheses involving the usage of pre-processing methods, wheat canopy segmentation methods, and whether the existing models from the past research can be adapted to classify wheat crop water stress. Hence, to construct an automation model for water stress detection, it was found that pre-processing operations known as total variation with L1 data fidelity term (TV-L1) denoising with a Primal-Dual algorithm and min-max contrast stretching are most useful. For wheat canopy segmentation curve fit based K-means algorithm (Cfit-kmeans) was also validated for the most accurate segmentation using intersection over union metric. For automated water stress detection, rapid prototyping of machine learning models revealed that there is a need only to explore nine models. After extensive grid search-based hyper-parameter tuning of machine learning algorithms and 10 K fold cross validation it was found that out of nine different machine algorithms tested, the random forest algorithm has the highest global diagnostic accuracy of 91.164% and is the most suitable for constructing water stress detection models.

# INTRODUCTION

Wheat is one of the world's most extensively consumed staple foods, accounting for roughly a quarter of total consumption. India is referred to as the "powerhouse of agriculture" worldwide because it has risen to the top of the worldwide milk, spice, and pulse production rankings. Furthermore, it is the world's second-largest wheat producer after China (*Dey, Dinesh & Rashmi, 2020*). According to statistics released by the Population Reference Bureau, the world's population is expected to have expanded by approximately 10 billion people by the end of 2050 (*Population Reference Bureau, 2021*), boosting the demand for agricultural products. Food security and sovereignty are the biggest challenges for many countries, especially in the wake of the circumstances emerging from the Russia-Ukraine conflict and climate change (*Ben Hassen & El Bilali, 2022*). Advancements in the field of high-throughput plant phenotyping and machine learning can help to overcome many of the challenges and streamline the process of identifying and classifying various biotic and abiotic stresses (*Komyshev et al., 2018*). Abiotic stress, such as drought, which is caused by a shortage of water, has been discovered to cause a considerable drop in wheat production by interfering with wheat crops'

Corresponding author
Ankita Gupta,
gupta89ankita@gmail.com

metabolism, growth, and yield. Due to below-average precipitation, droughts cause water shortages. El Niño occurrences and water abuse can produce droughts. Precipitation, soil moisture, and streamflow indicate drought. Crop growth simulation, water balance, and crop water stress index models (Landsat Soil Adjusted Vegetation Index (SAVI), Normalized Difference Vegetation Index (NDVI), and Landsat Enhanced Vegetation Index (EVI)) can identify wheat drought stress (*DROUGHT IN NUMBERS 2022— restoration for readiness & resilience, 2022*). Precipitation, temperature, solar radiation, soil moisture, evapotranspiration, and crop growth can be utilised to create and validate wheat drought models. A drought detection model should have accuracy, sensitivity, specificity, and missing data handling. Drought stress onset, duration, and severity vary by geography and climate. Certain factors can help a model detect drought in wheat images, for example leaf rolling, leaf senescence, leaf water potential, stomatal conductance and chlorophyll content can distinguish stressed from non-stressed crops. However, using meteorological, remote sensing, and soil moisture data can help to determine the crop's water stress status. Machine learning methods like Random Forest and Neural Networks can help find key traits and classify crops as stressed or unstressed.

Rainfed crops, such as wheat, are particularly vulnerable to unpredictable variations in the climate. Immediate efforts are required to address this issue before it worsens, preferably through the development of appropriate early stress detection systems. The timely detection of water stress in plants has become a matter of concern to avoid short-term income and yield losses as well as long-term consequences for rainfed farmers, which could lead to their abandonment of the agricultural profession. According to data from DEWS (drought early warning systems), around 42 percent of Indian land has been affected by drought as of January 1, 2019 (*van Ginkel & Biradar, 2021*; *Sharafi et al., 2021*). The Drought in Numbers, 2022 report, which was presented on May 11 at the UNCCD's 15th Conference of Parties (CoP15) (*DROUGHT IN NUMBERS 2022—restoration for readiness & resilience, 2022*), provides some significant information about the current drought situation and its effects on the Indian economy. Ever since the beginning of the twenty-first century, the frequency and extent of droughts are growing at an alarming rate all across the world. India comes under one of the severely drought-impacted countries, as contains a large portion of the world's drought-vulnerable regions. Drought affected nearly two-thirds of the nation from 2020 to 2022. Due to frequent droughts, India's Gross Domestic Product (GDP) declined by 2% to 5% between 1998 and 2017 (*DROUGHT IN NUMBERS 2022—restoration for readiness & resilience, 2022*). It is crucial to monitor and analyse drought's effects on wheat production because climate change is predicted to worsen drought stress in many locations. To reduce drought stress on wheat production, there is a need to breed drought-tolerant cultivars, use of precision irrigation, and improve water management. Drought stress affects 20–30% of the global wheat acreage, limiting productivity. Semi-arid and arid regions including the mediterranean, Middle East, Central Asia, and parts of Africa and Australia experience drought stress. Wheat has the largest water consumption throughout reproductive and grain-filling stages, making drought stress more likely. Drought stress can reduce wheat yield by 10–50%, depending

on severity, timing, and cultivar (*DROUGHT IN NUMBERS 2022—restoration for readiness & resilience, 2022*).

Plants can respond to drought stress in a variety of ways, and the major goal of our research is to understand each part of the wheat plant (*Komatsu & Hossain, 2013*) concerning water stress to find the responses and changes in different phenomena of the plant occurring due to stress using high throughput imaging technologies (*Abhinandan et al., 2018*; *Tounsi, Feki & Brini, 2019*). A deeper understanding of this can be accomplished by familiarising yourself with the full set of biological parameters that can be monitored with the help of computer algorithms (*Gupta, Kaur & Kaur, 2021*; *Schoppach et al., 2016*). Incorporating the latest phenomics and chlorophyll fluorescence technological advances into plant research can aid in the understanding and modelling of the various pressures that plants may encounter during their entire growth cycle. Both biotic and abiotic stress can be quantified and monitored using these advancements (*Sun et al., 2020*; *Tucci et al., 2018*). The study of re-emitted light emitted by a plant's body is one of the most efficient and straightforward methods of determining stress in plants. Light energy falling on a plant is diffused in three ways that are all equally important: first, photosynthesis, which is powered by light energy; second heat, which is released as a result of the dissipation of light energy, and third is chlorophyll fluorescence, which is a re-emission of light energy, all of which are equally vital (*Sánchez-Moreiras et al., 2020*). Wheat crop images are irradiated with photosynthetically active radiation (PAR) between 400 and 700 nm before performing remittance analysis of PSII (*Oxborough, 2004*; *Zhao et al., 2012*). The result of this process leads to the computation of chlorophyll fluorescence, which is a non-intrusive indicator of photosynthetic activity within the plant.

Fluorescence analysis is one of the simplest methods to determine stress as it accounts for 1–2 percent of total light energy and offers useful information on photosynthetic activity and energy loss in the form of heat within the plant body. The digital examination of the PSII photosystem can be performed using the fluorescence feature analysis of three-set images (fdark: null image, fmin: image with minimum fluorescence, and fmax: image with maximum fluorescence). The quantum efficiency of photosystem II can be calculated using the formula fv (variable fluorescence)/fmin, where fv equals the difference between fmax and fmin fluorescence values (*Kalaji et al., 2017*; *Sid'ko et al., 2017*). It refers to the plant's ability to adapt to stressful situations. Analysis of the fv/fm distribution patterns in the plant can quickly reveal the primary stress sources in the plant (*Gehan et al., 2017*). The development of machine learning models based on the properties extracted from plant images for detecting stress is a difficult task. This is due to the fact that identifying the most appropriate feature to map the ground truth requires a lot of domain knowledge, and at the same time, mapping concepts of the agricultural processes into the imaging process is hard (*Mantovani, Brito & Mantuano, 2018*; *Moya et al., 2019*). The next section explains the various options that contemporary researchers are applying to overcome such challenges.

## Survey methodology

A discussion on the latest research work done in the context of identifying water stress using image processing methods is given in this section. Contemporary literature gives

copious evidence that image-based analysis, which includes high throughput imaging feature analysis of plants, can be accomplished by applying a variety of functions to the pixels of the plant (*Vadakkenveettil, 2012*; *Löfstedt et al., 2019*). However, the image processing domain is not without bottlenecks. To overcome the challenge of the segmentation of plant body parts in plant images, *Kienbaum et al. (2021)* have used multiple preprocessing operations. For example, a linear or polynomial thresholding function may be applied to plant images to correctly identify shoot area, canopy temperature, and vegetation indices, among other things. The non-linear function analysis for extracting features from the images has been used in two ways: first, through the use of geometric statistics. Geometric statistics help estimate the height, convex hull, and centre of mass of the plant body parts. The geometric feature values of the plant that is suffering from some stress will be different from the healthy body part of the plant. The second is, of course, through the use of non-geometric descriptors, which can be used to perform biotic and abiotic stress analysis (*Dolferus et al., 2019*; *Oinam & Mehta, 2020*). For example, plant growth rate prediction, maturity, and yield prediction of wheat are the best indicators of some stress. These metrics have been used to make the detection of stress more precise, as it is quite challenging to have high precision agriculture equipment due to the role of multiple environmental factors. The attempt of many researchers to solve the key problems of stress detection is by conducting organ-level (leaf, stem, root, and canopy, *etc.*) analysis (*Chen et al., 2021*). However, from contemporary literature, it can be observed that there is not much agreement given on the kind of imaging features that suit best for the machine learning models, for the detection of various kinds of stress.

Although the algorithms for segmentation and machine learning are able to directly handle the input images, it has been noticed that the majority of image processing projects require some form of pre-processing operation. This is the reason that image preprocessing activities have an effect on the accuracy of subsequent algorithmic processes, including machine learning and segmentation. Therefore, the preprocessing algorithms are a necessary evil in the pipeline and scheme of things to design some kind of image processing system. This is because they improve the reliability of the image processing system in terms of their performance. The processes that make up preprocessing are designed to eliminate difficulties associated with low saturation, uneven aspect ratio, uneven brightness, and various sorts of noises (*Li & Xu, 2019*; *Jin et al., 2018*). It also helps to overcome problems that may occur due to incorrect camera calibrations and the presence of unwanted objects or artefacts in the image. These problems can be overcome with the help of this technique. At the same time, it is possible to find that resize functions that use interpolation techniques are used to improve and correct the aspect ratio of the images (*Zhang et al., 2022*). Current research in this field provides abundant evidence of the usage of several denoising methods such as median filter, non-local means filters (*Wu et al., 2018*), gaussian filter, total variation filter (*Caselles, Chambolle & Novaga, 2015*; *Gupta, Kaur & Kaur, 2021*; *Bose et al., 2016*; *Fernandez-Gallego et al., 2020*; *Gupta, Kaur & Kaur, 2022a*; *Hasan et al., 2018*; *Image Completion using Spiking Neural Networks, 2019*; *Pineda et al., 2017*; *Lazarević et al., 2021*; *Osroosh, Khot & Peters, 2018*; *Trivedi, Shukla & Pandey, 2022*; *Zeng & U, 2020*; *Zhi, Shi & Sun, 2016*), and bilateral filters to remove the

noise. In order to get rid of the shadows in the images, low-saturation areas, contrast enhancement techniques like Min-max and contrast stretching are usually utilised as pre-processing processes (*Trivedi, Shukla & Pandey, 2022*). Researchers have developed a number of different auto correction algorithms in order to improve the images' colour balance and achieve uniformity in the illumination and brightness. This will allow for greater control over how images are displayed.

Further, it can be observed from the current literature that *Zeng & U (2020)* have improved linear and non-linear contrast enhancement methods to obtain better segmentation results. Linear image improvement techniques including min-max stretching, thresholding function, and percentile stretching use contrast stretching. And nonlinear approaches like histogram equalisation, and gaussian stretch by analysing various pipelines in combination with denoising and enhancement gave the best pre-processing combination for wheat canopy images, which can greatly improve segmentation accuracy. This can be done using TV L1 denoising with a primal dual algorithms in combination with max contrast stretching enhancement procedures (*Gupta, Kaur & Kaur, 2022a*). By comparing wheat canopy and ear temperatures, wheat ear detection systems have been developed. *Gupta, Kaur & Kaur (2021)*, *Bose et al. (2016)*, *Hasan et al. (2018)*, *Image Completion using Spiking Neural Networks (2019)*, *Pineda et al. (2017)*, *Lazarević et al. (2021)*, *Osroosh, Khot & Peters (2018)* used the CLAHE method to improve image local contrast isolate neighbouring ears. Colour threshold segmentation was employed in HSV colour space to segregate canopy and ear high temperature disparities. To enrich images, sliding windows and CLAHE contrast adjustment techniques are used. Wheat ear counting images will have some noise due to the reflection of wheat leaves in sunlight, camera instability, and the natural surroundings. The wheat ear images needed to be enhanced and denoised using adaptive histogram equalisation and median filtering.

The colour and vigour of the plant deteriorate as a result of wilting. This can be seen visually and through the use of computer vision techniques (*Gupta, Kaur & Kaur, 2021*; *Osroosh, Khot & Peters, 2018*). In order to well distinguish between healthy and stressed plants classifiers such as support vector machine, artificial neural network has been trained with a specific set of characteristics that can characterise the histograms generated at the frequency of F520/F680, results of the analysis demonstrate the utility of multicolour fluorescence for plant phenotyping (*Pineda et al., 2017*). Absolute reflectance characteristics such as reflectance in red (RRed), green (RGreen), blue (RBlue), near-infrared (RNIR), and far-red (RFarRed) have also been utilised. *Lazarević et al. (2021)* have also employed hue (HUE), saturation (SAT), and value (VAL) as an alternative to visible reflectance for colour analysis. HUE takes into account the red, green, and blue colours, although it is displayed as a single channel with values ranging from 0 to 360 degrees. For every colour the saturation (SAT) indicates its intensity (pale or intense colour), while the value (VAL) indicates whether the colour is bright or dark. According to the results of colour analysis, reflectance in red, green, and blue dramatically increased if there is an extreme drought. Hence, drought significantly affect the colour reflectance by boosting the

intensities in RRed, RGreen, and RBlue bands substantially. HUE2D is another parameter that has higher sensitivity to drought.

Where comprehensive domain expertise was not available, *Hasan et al. (2018)* used region-based convolutional neural networks (R-CNN) for labelling different types of stresses on the plant images. Since there are limited dataset resources for plant images in the context of wheat. There is an urgent need to build systems that can annotate images without the help of domain experts. *Bose et al. (2016)* and *Image Completion using Spiking Neural Networks (2019)* trained a system that annotated the wheat dataset "SPIKE" to predict grain yield. A collection of images was taken at various growth phases of wheat to analyse spikes and estimate yield, with an average detection accuracy ranging from 88 to 94 percent for the various models. To reduce the genotype-phenotype gap, plant phenomics is the most efficient approach that has been employed to date (*Pasala & Pandey, 2020*; *Gjuvsland et al., 2013*). *Ghosal et al. (2018)*, *Gao et al. (2020)* working in the field of wheat phenomics employ deep plant phenomics platforms that are based on Convolution Networks technology. These platforms can automate the process of phenotyping by providing accurate and efficient phenotypic measurement. It can be further observed that the use of CNN for abiotic and biotic stress detection in plants has been demonstrated to be more effective than other current computer vision technologies, such as deep learning (*Zhou et al., 2021*). To obtain efficient comparative results, *Kamarudin, Ismail & Saidi (2021)* have run a competition among a variety of deep learning models in the context of detection and classification. According to experimental findings, Google Net outperformed Alex Net and Inception V3 in terms of accuracy and error rate, with an accuracy score of 98.3 percent and an error rate of less than 7.5 percent to classify plant stresses (*Chandel et al., 2020*). Further, *Santos et al. (2021)* worked on optimising classifiers for water stress detection in wheat crops, in which UAV aerial RGB images are segmented for vegetation extraction using vegetation index thresholding.

Support vector machine (SVM) classifiers (*Su et al., 2020*) were trained using features extracted from images, which was further optimised using a Bayesian optimizer to improve the performance of the classifier (*Elvanidi et al., 2017*). It was found that when only spectral intensities were used, the optimised classifier achieved an accuracy of 89.9 percent with an F1 score of 87.7 percent, and from this outcome it was inferred that the accuracy can be improved further by combining spectral intensity features with colour index features. This ultimately led to an accuracy of 92.8 percent with an F1 score of 91.5 percent. A supervised learning approach called a gradient boosting decision tree utilised fourteen colour and texture features for efficient classification. The implemented method exhibited a successful detection performance between control and water stress conditions in the maize fields. According to the results of this article (*Zhuang et al., 2017*), the recognition accuracy of three water treatments was 80.95% and the accuracy of water stress reached 90.39%. To determine the most accurate algorithm for identifying droughts, the classification and prediction capacities of decision tree (DT), genetic programming (GP), and gradient boosting decision tree (GBT) algorithms have been examined in both the testing and training phases. It was observed that GP models with scaled sigmoid functions at their roots are remarkably good at classifying and forecasting drought (*Mehr, 2021*).

*Sun et al. (2019)* have also employed a time-series analysis of chlorophyll fluorescence (ChlF) to analyse the ChlF fingerprints of salt overly sensitive (SOS) mutants under drought stress. Sparse autoencoders (SAEs) neural network, a time-series deep learning approach, was utilised to extract time-series ChF features, which were used in four classification models including linear discriminant analysis (LDA), k-nearest neighbour classifier (KNN), naïve Bayes (NB) and support vector machine (SVM). According on the findings, the LDA classification model's discrimination accuracy was found to be 95%. According to the findings (*Xia et al., 2022*), the induction curve contains crucial information about plant physiology. This can be justified by analysing the results obtained from SVM classifier, can classify the severity of drought stress more accurately than the KNN and Ensemble, with a classification accuracy of 86.7 percent for the induction curve as input compared to 43.9 percent for Fv/Fm and 72.7 percent for induction characteristics.

Photosynthesis-based kinetics analysis of photosynthetic traits, such as PSII quantum yield (fv/fm), Fmin (minimum fluorescence), and Fmax (maximum fluorescence) (*Botyanszka et al., 2020*), can capture the effects of climatic variation on the photosynthetic activity of the plant. A valuable tool for understanding how damage develops and how responses are organised in crops can be developed using the chlorophyll fluorescence imaging approach. The chlorophyll fluorescence images are captured using CCD (camera) (*Xu et al., 2021*). These cameras experience a variety of noises, including thermal, white, dark current, reset, flicker, and amplification noise. The hardware circuitry and post-processing algorithms of the camera handle almost all sorts of noise. However, it was found that some random noise does enter the chlorophyll fluorescence images as a result of these factors, and at the same time, the problems with illumination variations, camera calibration errors, and various settings under which the images were obtained also contribute to some proposition of noise in the images leading to reduced SNR ratio. Due to this extraction of plant's organ such as wheat canopy become a difficult task and additional pre-processing operations are required which has been already implemented by the (*Gupta, Kaur & Kaur, 2022a*).

## MATERIALS AND METHODS

This section explains the materials and methods used for achieving the goals of this study. The dataset of chlorophyll fluorescence images were obtained from a public repository (*Gupta, Kaur & Kaur, 2022c*; *Sandhu, 2019*) and is of the Raj 3765 wheat variety. This wheat variety is most predominantly sown in the North Western Plain Zone (NWPZ). The collection of 24 images per day are collected over 60 days (vegetative growth stage) for each experiment (control and drought) is represented as 2,880 (1,440 Control and 1,440 drought) instances with the resolution of a 72-dot-per-inch RGB image collection. The wheat plants were grown in pots in laboratory conditions and one set of pots was not given water to induce water stress in the plants. The analysis contains a spatio-temporal difference between the periods of Fmin (minimum fluorescence) and Fmax (maximum fluorescence). The data gathering procedure involved the application of a visible light

spectrum-based colour camera to capture images of the plant canopy from the top or plan view.

## Pre-processing and preliminary study

In the course of the intended experiments, it was found that contrast enhancement and random noise removal are prerequisite requirements for chlorophyll fluorescence (CF) images before they are fed to any segmentation algorithm to increase the segmentation accuracy. Because the raw chlorophyll fluorescence wheat image is never smooth, it consists of random graininess that is strong enough to conceal fine details. It has been empirically demonstrated that the contrast stretching (min-max) method of contrast enhancement when combined with the TV-L1 denoising with a Primal-Dual algorithm is the most effective pre-processing technique for identifying the image region with the highest level of photosynthetic activity (*Gupta, Kaur & Kaur, 2022a*). The Primal-Dual algorithm removes image noise using total variation (TV) and L1 regularisation. TV measures image fluctuation mathematically. It is useful for denoising images since it preserves edges and details while reducing noise. L1 regularisation adds a penalty term to an optimization problem to encourage modest absolute values. Because it preserves an image's sparsity (few non-zero values), it is used for denoising. Primal-Dual solves convex optimization issues. It solves PDEs and other non-smooth problems well. Primal-Dual TV-L1 denoising iterates. It starts with a denoised image estimate and iteratively updates it by solving subproblems that minimise the TV and L1 regularisation terms. Stopping criteria stop the algorithm (*e.g.*, when the difference between two consecutive estimates is below a certain threshold). It removes noise from grey-scale and colour photos while preserving fine details and edges. It can be observed (see. Fig. 1) that six pipelines were used to identify which pre-processing method would yield and support the wheat canopy segmentation. The purpose of the methods was to overcome the problems of low contrast and noise. Hence, denoising and contrast enhancement methods were put to test and it was found that TV-L1 denoising with a Primal-Dual algorithm and min-max contrast stretching is the best suited as they preserve the texture property of an image. Hence, the pipeline II (see Figs. 1D–1F) was selected for further research workflow. The code and image outputs data from this research work were deposited in a public repository (*Gupta, Kaur & Kaur, 2023*) to enable future research work.

For the precise extraction of the region of interest (ROI), wheat canopy segmentation experiments were conducted to evaluate seven segmentation strategies, viz., global static thresholding, global automatic thresholding (Otsu), mean shift, edge detection operators, k-means (based on four means), watershed, and the "Cfit K-means algorithm" (*Gupta, Kaur & Kaur, 2022b*). The IOU (intersection over union) metric score has been used for the validation of the segmentation of regions of interest (wheat canopy). From the results, it has been observed that the Cfit K-means algorithm provides the highest IOU score of 95.75 with pre-processing and 59.8 without pre-processing, among the seven segmentation algorithms implemented and investigated (*Gupta, Kaur & Kaur, 2022a*). The pre-processed images prove to be fruitful in enhancing the segmentation accuracy by 36%, segmentation results are validated using the IOU score (refer to Table 1), For the sake of

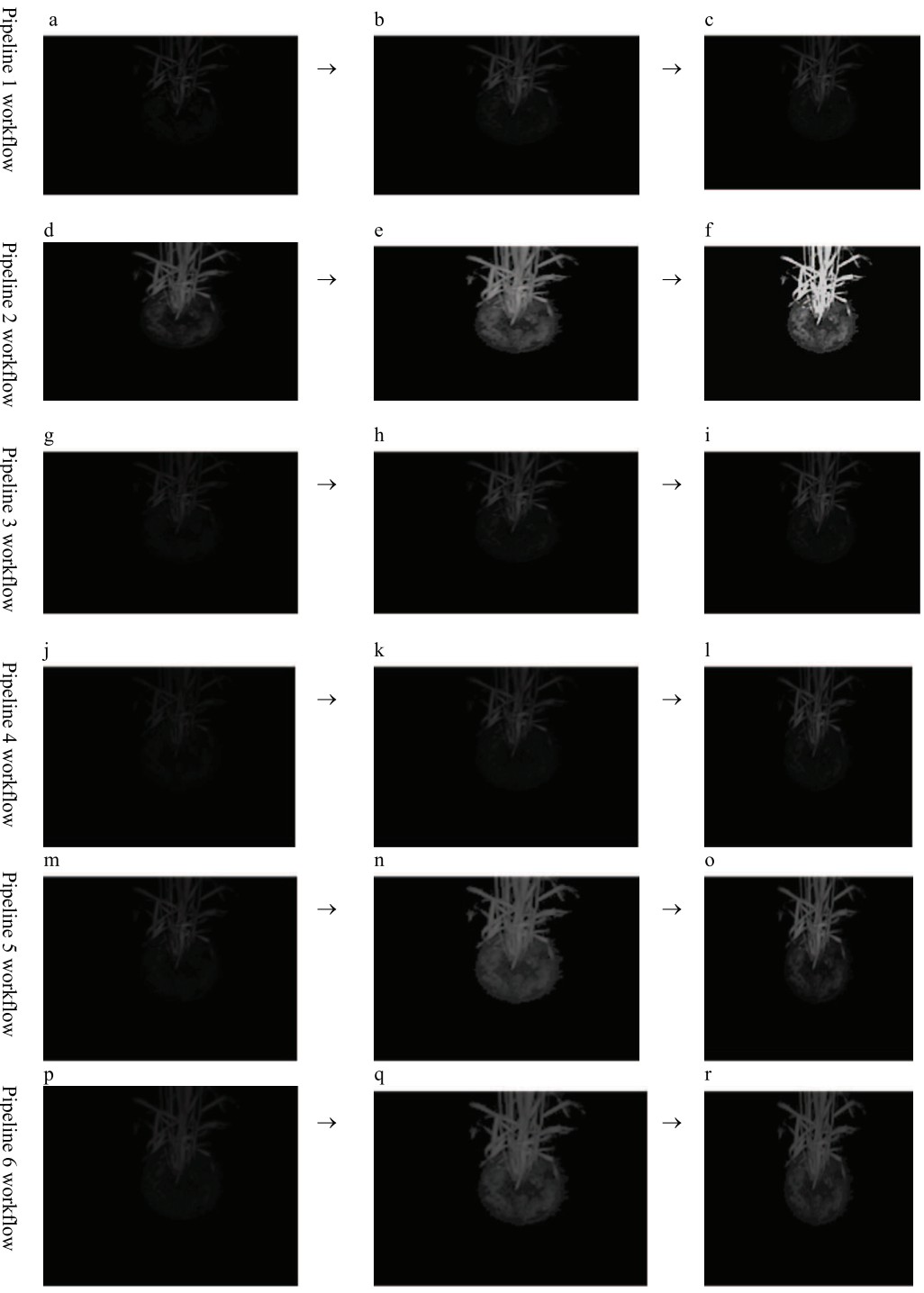

**Figure 1 Comparative Analysis of pre-processing methods workflow.** (A) Noise removal technique used: non-local means denoising. (B) Contrast enhancement technique used: contrast stretching (Min-Max). (C) Output preprocessed image following pipeline 1. (D) Noise removal technique used: TV-L1 denoising with a Primal-Dual algorithm. (E) Contrast enhancement technique used: contrast stretching (Min-Max). (F) Output preprocessed image following pipeline 2. (G) Noise removal technique used: non-local means denoising. (H) Contrast enhancement technique used: CLAHE. (I) Output preprocessed image following pipeline 3. (J) Noise removal technique used: TV-L1 denoising with a Primal-Dual algorithm.

**Figure 1 (continued)**
(K) Contrast enhancement technique used: CLAHE; (L) Output preprocessed image following pipeline 4.
(M) Noise removal technique used: non-local means denoising. (N) Contrast enhancement technique
used: histogram equalization. (O) Output preprocessed image following pipeline 5. (P) Noise removal
technique used: TV-L1 denoising with a Primal-Dual algorithm. (Q) Contrast enhancement technique
used: histogram equalisation. (R) Output preprocessed image following pipeline 6.

**Table 1 Comparative analysis of CFitk-means algorithm with and without pre-processing.**

| Segmentation algorithm | Sample size | | | | Average IoU score |
|---|---|---|---|---|---|
| | 25 | 50 | 75 | 100 | |
| CFitk-means (without pre-processing) | 0.59 | 0.60 | 0.612 | 0.61 | 0.60 |
| CFitk-means (with pre-processing) | 0.96 | 0.95 | 0.96 | 0.97 | 0.966 |

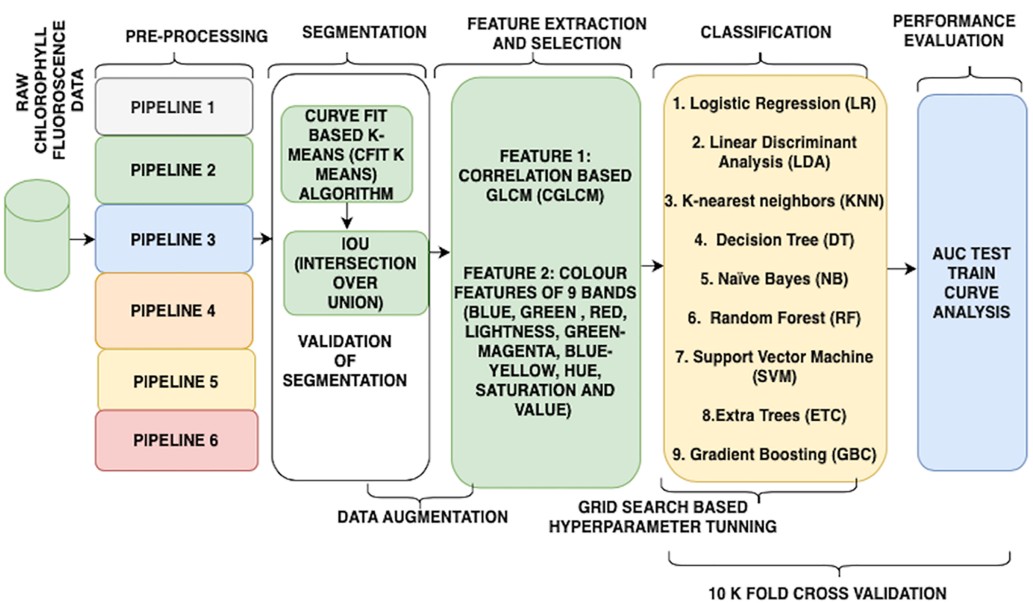

**Figure 2 Flow of the research.**     

reproducibility and portability, data and code linked to the pre-processing pipeline and
segmentation are freely available (*Gupta, Kaur & Kaur, 2022d*). This will aid in the
creation of a dataset for the automated detection of water stress using machine-learning
algorithms and other techniques.

Further, initial experiments and tinkering in this context show that the use of
correlation-based gray-level co-occurrence matrix (CGLCM) and colour features (colour
proportion of pixels in each nine bands) is fairly suitable for this purpose. Therefore, this
research work aims to identify the most suitable workflow to accomplish the task of
classification. For better understanding (refer to Fig. 2). The next section explains the
implementation and the construction of the wheat water stress detector.

## Feature extraction and selection

The raw image of chlorophyll fluorescence modality has been employed in this research as a beginning point for creating the mentioned water stress automation model. The images have been acquired at the particular wavelength and frequency when the excitation of the chlorophyll fluorescence occurs and stops. Thus, wheat CF images' pixel colours fluctuate throughout. The dataset's file structure shows that each PSII activity cycle has 24 images per day for both drought and control for 60 days, starting with non-excitation and ending with fluorescence excitation. The difference in the proportion of pixels of each colour band gives a clear-cut signal about the health of the wheat plant. Hence, a total of five features: chlorophyll fluorescence (PSII), texture/GLCM, morphological/shape features, correlation-based features, and colour percentage of various nine bands (blue, green, red, lightness, green-magenta, blue-yellow, hue, saturation and value) are extracted from plant images and submitted to comparison analysis in order to construct an automatic ML-based identification model. In the comparative study, all of these features were determined utilising a backward elimination process for feature selection. The criteria for elimination were based on the global diagnostic accuracy metric of the machine learning models. If a subset of characteristics gave a global diagnostic accuracy greater than 80% (referred to as baseline accuracy) during testing, that subset was considered for inclusion in the wheat stress detection model for further investigations on its quality of performance.

The observations from these operations show that correlation-based GLCM and colour proportion values of nine bands ((blue, green, red, lightness, green-magenta, blue-yellow, hue, saturation, and value) as features) in combination provide the most reliable information about changes that happen when the plant is under stress. So, this combination has been selected and rest all features has been eliminated. The 23 GLCM metrics (autocorrelation (autoc), contrast: (contr), correlation (corrm), correlation (corrp), cluster prominence: (cprom), cluster shade (cshad), dissimilarity (dissi), energy (energ), entropy (entro), homogeneity (homom), homogeneity (homop), maximum probability (maxpr), sum of squares (sosvh), sum average (savgh), sum variance (svarh), sum entropy (senth), difference variance (dvarh), difference entropy (denth), information measure of correlation1 (inf1h), information measure of correlation2 (inf2h), inverse difference (INV) is homom (homom), inverse difference normalised (INN) (indnc), inverse difference moment normalised (idmnc)) are used to obtain constructive information regarding water stress in the wheat plant. Texture/GLCM analysis indicates "change" due to water stress in the wheat canopy. The current work draws its methodology in the context of GLCM operations from the article. The Kendal formula has been utilized for computing correlation among 23 GLCM features (see. Fig. 3).

Only correlation-based GLCM (CGLCM) feature selection yielded to over-fitted models as almost all the attributes had some correlation with each other. However, CGLCM combined with nine colour band features provided a better feature set. The morphological/shape features or integral geometrical features such as (eccentricity, area, perimeter, and convex hull) made no significant change in achieving accuracy above the baseline accuracy of 80%. Hence, they were dropped from the list of features when correlation-based GLCM

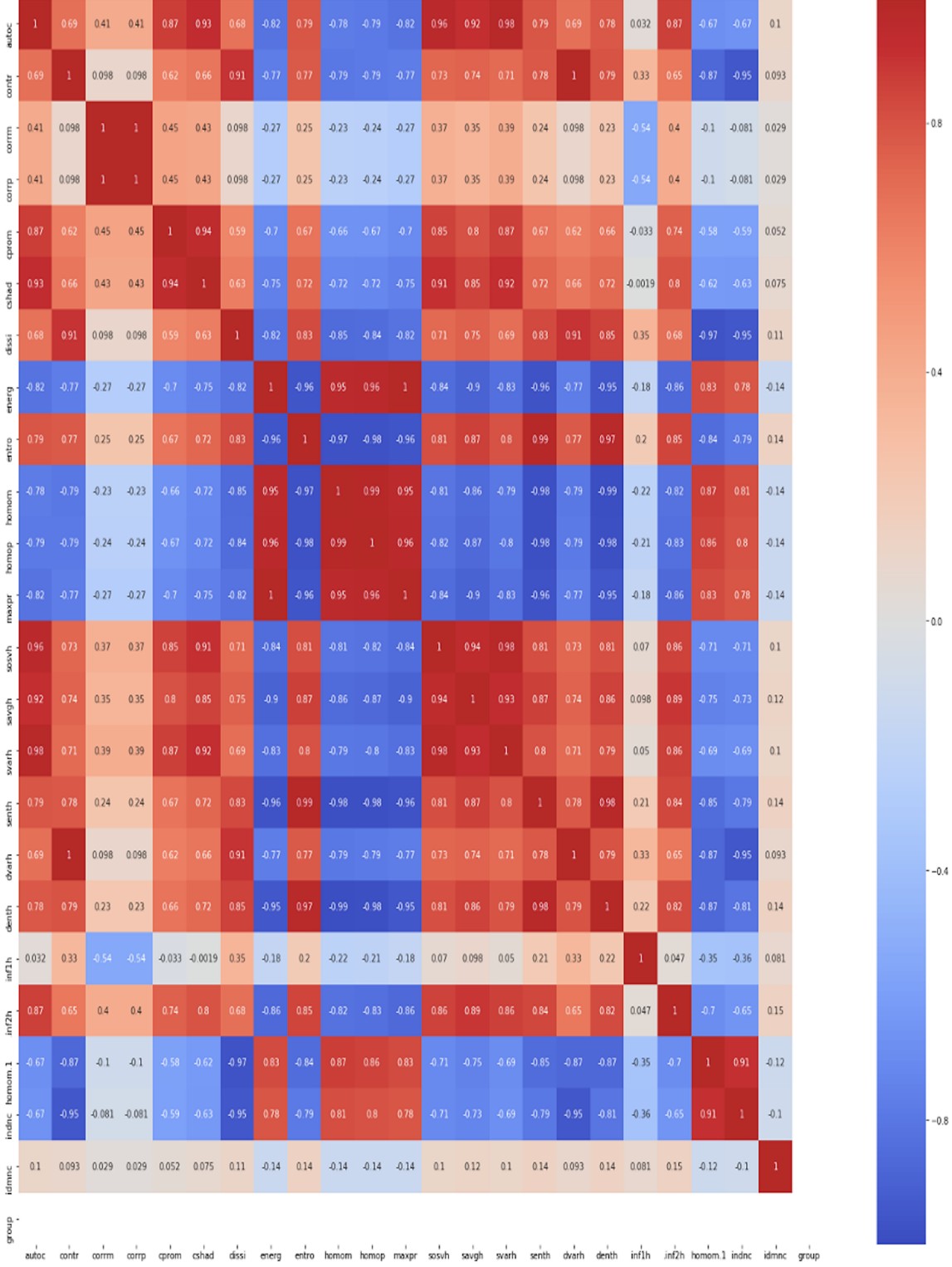

**Figure 3 Kendall correlation analysis for 23 GLCM features.**

**Table 2 Comparative performance analysis of the machine learning algorithms.**

| S. no | Algorithms | AUC test | AUC train | Diff (Test-Train) |
|---|---|---|---|---|
| 1 | NB | 0.74597268 | 0.74749576 | −0.0015231 |
| 2 | LDA | 0.74467457 | 0.74880706 | −0.0041325 |
| 3 | LR | 0.8053969 | 0.81098665 | −0.0055898 |
| 4 | SVM | 0.81517629 | 0.87208612 | −0.0569098 |
| 5 | GBC | 0.81517629 | 0.87208612 | −0.0569098 |
| 6 | KNN | 0.72575046 | 0.81187261 | −0.0861221 |
| 7 | RF | 0.91164728 | 1 | −0.0883527 |
| 8 | ETC | 0.90773481 | 1 | −0.0922652 |
| 9 | DT | 0.8723432 | 1 | −0.1276568 |

or CGLCM and colour features in combination were given as input to machine learning models, this led to a substantial increase in the accuracy of the classification algorithms (refer to Table 2). The performance was tested using nine classifiers, as demonstrated in the next section.

## RESULTS

Multiple machine learning methods for the automatic diagnosis of drought water stress have been investigated. For every experiment the ratio of the training dataset was kept at 80% of the total dataset (2,880) and 20% dataset was used as testing dataset. However, it must be noted that during the process of 10 K fold cross validation, the dataset was randomly divided into k groups (in our example, k = 10) of nearly equal size. The first fold is used as a validation set, while the following k-1 folds are used to fit the procedure.

A rapid prototyping python library lazy predict was deployed and 20 models of different machine learning algorithms were generated. For further analysis, the top nine machine learning algorithms, viz., logistic regression (LR), Linear Discriminant Analysis (LDA), K-nearest neighbors (KNN), decision tree (DT), naïve Bayes (NB), random forest (RF), support vector machine (SVM), extra trees (ETC), and Gradient Boosting (GBC), were selected for further analysis.

The algorithms were tested for the single wheat variety "RAJ 3765". However, due to a small dataset issue, data augmentation was done to increase the size of the dataset 10 times its original size. During data augmentation, we randomly augment the dataset by rotating images at different angles and flipping images horizontally/vertically. The performance analysis of the machine learning models indicates that the RF algorithm appears to be the most accurate as per the statistics given in Table 2.

The results are validated using 10-fold cross-validation for all the algorithms and fine-tuned using the grid search algorithm at all stages. From this, it can be concluded that the pre-processing step plays the most predominant role in building the automatic detection model. Dependency of the other steps, depending upon the correctness and quality of the image achieved after this step. From Table 3, it can be observed that tree-based algorithms are demonstrating a high level of performance. However, the diagnostic efficacy of the

**Table 3 Hyper parameters used in respective algorithms to fine tune the model's performance.**

| S. no | Algorithms | Hyper parameters found by grid search method |
|---|---|---|
| 1 | NB | # of Class = 2; # of attributes = # of feature rows; normalization = min_max; probability of each class = 0.5; variable smoothening = 1 |
| 2 | LDA | # of Class = 2; Solver = lsqr; tol = 1; shrinkage = auto |
| 3 | LR | # of Class = 2; Penalty = L2; tol = 1; c = 0.98; solver = lbfgs; class_weight = balanced; multi_class = ovr; max_iterations = 100 |
| 4 | SVM | # of Class = 2; Kernel = poly; C = 4.5; gamma = 0.01 |
| 5 | GBC | # of Class = 2; Loss = log_loss; learning rate = 0.5; maximum_depth = 3; n_of estimators = 50; criterion = mse; samples_split = 2; max_features = auto |
| 6 | KNN | # of Class = 2; Algorithm = auto; n_neighbours = 5; leaf_size = 20; weights = uniform; metric = minkowski |
| 7 | RF | # of Class = 2; Criterion = Gini; splitter = best; max_depth = 7; min_samples_split = 10; min_sample_leaf = 2; min_wt_fraction = 0; max_features = (no. of samples); class_weight = balance |
| 8 | ETC | # of Class = 2; Criterion = Gini; splitter = best; max_depth = 5; min_samples_split = 10; min_sample_leaf = 1; min_wt_fraction = 0; max_features = (no. of samples); class_weight = balance |
| 9 | DT | # of Class = 2; Criterion = Gini; splitter = best; max_depth = 3; min_samples_split = 10; min_sample_leaf = 1; min_wt_fraction = 0; max_features = (no. of samples); class_weight = balance |

classifier in terms of water stress detection needs deeper analysis with the help of AUC ROC curve analysis (see. Fig. 4). The curves drawn here are used to evaluate the effectiveness of machine learning algorithms.

The AUC ROC is a visual approach for understanding the binary classifier's diagnostic capabilities (called global diagnostic accuracy). These curves are used to analyse the effectiveness of the machine learning algorithms, which have numerous advantages. One of these advantages is that it is more accurate to conclude which algorithm is reliable and stable. There is no effect of scale on AUC. It is not concerned with the absolute values of the forecasts themselves but rather with how well they are ranked. The AUC is not affected by the threshold for classification. Regardless of the value of the classification threshold that is selected, it aggregates the performance of the classifier across all the possible thresholds and then evaluates the prediction accuracy of the model. Both the testing and training datasets have the same distribution, and there is no covariate shift or drift in the features, according to a brief investigation of the feature distribution that was performed between the two datasets. It has been noted that altering the number of independent variables allows us to obtain various interpretations of the same automation model hence, all these models have been tested using a 10 K-fold validation process.

There is not a single case in which the AUC is close to zero in the testing phase from the beginning and nor is there a categorical switch in the classes due to this fact. At the same time, it can also be observed that there are no absolute results where positive class instances were predicted with full accuracy in the final testing phase with the threshold of 0.80. This is because there are no absolute results where positive class instances were predicted with full accuracy and match.

When compared to the AUC values of the training phases, it can be seen that the values of AUC drop significantly during the testing phases in almost every scenario. However, the

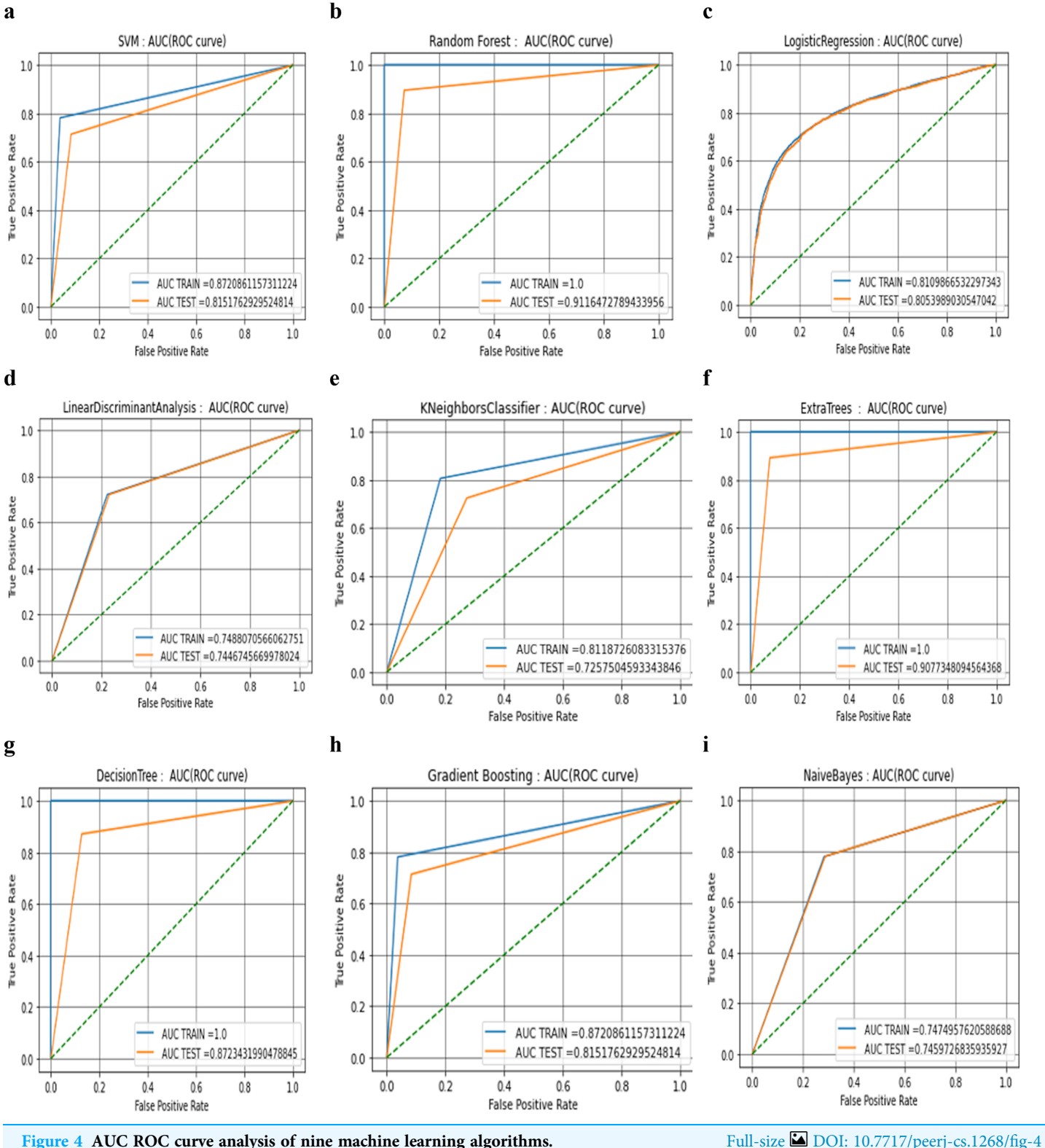

**Figure 4 AUC ROC curve analysis of nine machine learning algorithms.**

decline is greatest in the cases of the DT algorithm (−0.1276568) (see. Fig. 4G) and the ETC algorithm (−0.09226519) (see. Fig. 4F) followed by RF algorithm (−0.0883527) (see Fig. 4B). Because of this, it appears that three of these models have a significant risk of producing results that are an excessive fit to the data. At that time, the distance between the curves of the training and the testing was measured, and it was discovered that the distance between the curves in graphs of four classifiers among nine tested classifiers (K Nearest Neighbours (see. Fig. 4E), Extra trees (see. Fig. 4F), Random Forest (see. Fig. 4B), and Decision Trees (see. Fig. 4G)) is slightly more than expected. If the gap is larger, it indicates that there is a need to further improve the process of training, which may be done by either adding more cases or optimising the parameters on which the algorithm is based. A larger gap between the testing phase and the training phase could potentially be a symptom of too many noisy features that prevent algorithms from learning from fresh examples. Because these results are produced by hyper-parameter tuning that was carried out with the assistance of the grid search method, further optimization of the algorithm was not possible, and it was advisable to investigate additional methods such as data augmentation and data fusion instead. It is important to notice that the data augmentation process has already been utilised, and further hungriness of the learning algorithm cannot be permitted because it may lead to results that are only cosmetic.

From the outcome of the ROC-AUC curve analysis, it was observed that algorithms that have training and testing curves that overlap with each other include the logistic regression (see. Fig. 4C), naïve Bayes (see. Fig. 4I), and linear discriminant algorithms (see. Fig 4D). This indicates that the naïve Bayes algorithm has a minimal amount of training loss error and the algorithm is capable enough to interpret the data patterns in both phases of the process. From the table it can clearly be inferred that the smallest difference between the values of testing and training phase belongs to the NB algorithm (−0.00152308) followed by LDA and LR algorithms respectively (see. Table 2). The performance drop shown by SVM and GBC (see. Fig. 4H) are almost similar and have difference values that are intermediate (−0.0569098) (see. Table 2).

Because the AUC ROC graphs for decision trees, random forest, and extra-tree methods are practically perfect during the training phase, but the values drop significantly during the testing phase, this implies that almost every member of the tree-based family of algorithms has some degree of variability in their performance. The gradient boosting approach does not have a perfect AUC ROC curve in the training phase.

The true positive rate of the SVM initially climbs to 0.8 and then goes maximum to 0.87 in the training phase, but its value also lowers in the testing phase (see. Fig. 4A). Nevertheless, it is important to note that the form of the curves in both phases is practically identical. In the case of (LR, LDA, and KNN), the shape of the curves in both phases is virtually identical.

The tree algorithms RF, ETC, and DT are able to give us a higher degree of accuracy and are superior in terms of other performance measures such as AUC test-train accuracy, precision, recall and F-score. RF is the tree method with the highest test accuracy and the smallest gap (−0.0883527) (see. Table 2) between the training and testing curves. As a result, it appears that this approach is the most appropriate one that should be used for

determining whether or not there is water stress. In addition to this important aspect of the RF method, you should take into account the fact that it can automatically manage multicollinearity because it separates the variable into a tree structure before processing it. As a result, there is a lower chance of the model being over-fit to the data.

## DISCUSSION

The purpose of the research as a whole was to develop an automated system for detecting water stress in RAJ 3765 wheat. In this study, we show that previous research has helped us overcome obstacles such as the need to use invasive technologies to quantify the water stress experienced by wheat crops. The workflow of this research is informed by a number of hypotheses concerning the use of pre-processing methods, wheat canopy segmentation methods, and whether the present models from prior work can be adapted for classifying wheat crop water stress. These hypotheses concern the use of numerous methods concerning the use of pre-processing methods.

The small volume of the dataset is the biggest barrier which is overcome using data augmentation, which led to an improvement in the accuracy of water stress identification. In raw wheat canopy CF images, random noise caused by thermal activity of photons and insufficient saturation made it difficult to make out the wheat canopy boundaries. Therefore, TV-L1 denoising using the Primal-Dual approach resulted in the greatest improvement to the segmentation accuracy of the C fit k-means algorithm. The segment pixels' gradient difference from the edges is increased as the contrast is stretched. This research was wrapped up with a contest featuring nine different machine learning models. A nine-machine learning model competition concluded this research. The competition and a comparative study sought to build the best machine model. Grid search and 10 K fold cross validation helped build a trustworthy Random Forest water stress detection automation model. This study will increase food supplies, saving lives. It will improve agriculture and many livelihoods. This work will build an image processing method to quantify drought stress on Indian wheat variety. It will then use classifiers to automate drought stress detection, making crop stress status assessment fast and more accurate.

## CONCLUSIONS

Multiple cases and hypotheses concerning the use of pre-processing methods, wheat canopy segmentation methods, and whether the present models from prior work can be adapted for categorising wheat crop water stress inform the workflow of this research. Consequently, it was determined that the most effective pre-processing operations for constructing an automation model for water stress detection are TV-L1 denoising with a Primal-Dual algorithm and min-max contrast stretching. Using the IoU measure, the curve fit K-means method was verified for the best accurate segmentation of the Wheat canopy. For automated water stress monitoring, fast prototyping of machine learning library suggested that just nine models need to be investigated. After thorough grid search-based hyper-parameter tweaking of machine learning algorithms and 10 K fold cross validation, it was determined that, out of the nine machine algorithms evaluated, the random forest approach is the most suitable for building water stress detection models.

The results of the tests indicate that a comprehensive assessment of nine machine learning algorithms provided sufficient data to support the conclusion that the random forest algorithm is the most suitable technique for water stress detection. Colour + correlation-based GLCM parameters in conjunction with image processing have been described for the first time in the literature for the identification of water stress. One of the most significant concerns observed in the raw chlorophyll fluorescence wheat canopy images was that they contained a certain amount of instrumentation-caused noise, and many shots exhibited low saturation, making the edges of the wheat canopy difficult to discern. Consequently, different noise approaches and contrast improvement methods were applied to increase the overall image quality, and it was shown that the TV-L1 denoising with a Primal-Dual method is the most effective for improving the segmentation accuracy of the Cfit k-means algorithm. This is done utilising the results of the contrast stretching min-max method increasing the gradient difference between pixels within segments and pixels on the segment's edges.

KNN and gradient boosting were discovered to be the algorithms most susceptible to incorrect classifications. As a result, more investigations were conducted. During the initial phase of learning, the gap between the training and testing curves for four of the nine analysed classifiers (K nearest neighbours, extra trees, random forest, and decision trees) was slightly more than anticipated. If the discrepancy is bigger, training must be enhanced by adding more examples or adjusting algorithm settings. An excessive number of noisy characteristics may impede an algorithm's ability to learn from fresh examples, resulting in a wider gap between the testing and training phases. Since these outcomes are the result of optimising grid search hyper-parameters, further optimization was not possible.

The data augmentation technique has already been implemented, and the algorithm's insatiable appetite may have aesthetic repercussions. The training and testing curves for logistic regression, naïve Bayes, and linear discriminant algorithms overlap, as determined by ROC-AUC analysis. This illustrates that the logistic regression technique has a small training loss error and can identify data trends in both phases. Almost every tree-based algorithm exhibits overfitting, since the AUC ROC graphs for DT, RF, and extra-trees approaches are nearly flawless during training but significantly degrade while testing. During the training phase of gradient boosting, the AUC ROC curve is imprecise. This restriction is algorithmic in nature. During training, the true positive rate of the SVM improves to 0.80 and 0.87 before declining. The shapes of both levels are nearly identical. In both stages, the (LR, LDA, and KNN) curves are virtually identical. In the last stage of SVM training, the decreases in values during training are accounted for, and TPR reaches a maximum of 0.87, which is more than the testing maximum of 0.81. RF has the lowest gap (−0.0883527) between the testing and training curves and highest accuracy. Therefore, the random forest algorithm is most appropriate for detecting water stress as it is able to deal with multicollinearity automatically by splitting variables into a tree for processing. Consequently, underfitting and overfitting is less likely. The proposed study will enhance agriculture and a variety of livelihoods. By using an image processing technique to measure the impact of drought on Indian wheat variety. In order to quickly and accurately assess

crop water stress status, it will then automate the identification of drought stress using comparison between the classifiers.

### Funding
The authors received no funding for this work.

### Competing Interests
The authors declare that they have no competing interests.

### Author Contributions
- Ankita Gupta conceived and designed the experiments, performed the experiments, analyzed the data, performed the computation work, prepared figures and/or tables, authored or reviewed drafts of the article, and approved the final draft.
- Lakhwinder Kaur conceived and designed the experiments, performed the experiments, analyzed the data, prepared figures and/or tables, authored or reviewed drafts of the article, and approved the final draft.
- Gurmeet Kaur conceived and designed the experiments, analyzed the data, prepared figures and/or tables, authored or reviewed drafts of the article, and approved the final draft.

### Data Availability
Gupta, ankita; Kaur, Dr Gurmeet; Kaur, Dr. Lakhwinder (2022), "Pre-processing of drought/water stress detection chlorophyll fluorescence wheat images for efficient segmentation", Mendeley Data, V3, DOI 10.17632/4y8s925fkm.3.

Gupta, ankita; Kaur, Lakhwinder; Kaur, Gurmeet (2022), "Automatic Water Stress detection in wheat crop canopy using Chlorophyll fluorescence image dataset", Mendeley Data, V2, DOI 10.17632/2mpd7d3vry.2.

Gupta, ankita; Kaur, Dr. Gurmeet; Kaur, Dr. Lakhwinder (2022), "ROI extraction of chlorophyll Fluorescence wheat canopy images using novel Curve Fit Based K-means segmentation Algorithm for automatic drought stress detection using machine learning ", Mendeley Data, V4, DOI 10.17632/crb5tkbvpb.4.

Sandhu, Sukhjit (2019), "Plant Stress Analysis Based on Chlorophyll Fluorescence and Image Processing", Mendeley Data, V2, DOI 10.17632/jnjd835ncg.2.

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
