# Peer review of "Drought stress detection technique for wheat crop using machine learning"

_PeerJ Computer Science, doi:10.7717/peerj-cs.1268_

## Round 0.1 · original submission · Major Revisions

The authors are asked to overcome the suggested comments.
Also please enhance the literature survey with recent papers to justify the preprocessing of the images and why only a certain kind of preprocessing is used.

Reviewer 1 ·

Basic reporting

Gupta et al., conducted an exploratory study using machine learning algorithms to determine the most suited algorithm for detecting water stress in wheat utilizing canopy images. Chlorophyll fluorescence is an indicator of a plant’s status. Any internal or external perturbations have a direct impact on chlorophyll fluorescence. Under drought to conserve water, plants minimize their photosynthetic activity and show a reduced chlorophyll accumulation and hence fluorescence. Application of machine learning using canopy image datasets generated by hyperspectral imaging can be beneficial for large fields where farmers can get early information of water stress which can be managed timely to avoid crop losses. Authors have tested eight different machine learning algorithms, where they concluded that Support Vector Machine algorithm was most suitable for constructing water stress detection models.
The work is timely, well thought and executed; however, there are some concerns that need to be addressed before the acceptance of this manuscript.

Concerns:
1. Overall, language has a lot of issues, and the flow is missing. There are formatting errors, and different fonts are used.
2. Comparative analysis of diverse deep learning approaches in lines 150-160 should be summarized in tabular form. That would be easy for readers.
3. Material and Method section is not appropriately written.
Line 231: “The data gathering procedure will involve….”. it should be ‘The data gathering procedure involved….”
Line 233: Dataset was created and deposited in a public repository to enable future replications. Where was the dataset submitted? Please provide link for the repository.
Line 234: “Both the dataset and code used to create the machine learning model are freely available.” Which code and dataset? What’s the source? Wasn’t’ the dataset created by the authors themselves in this study?
Whole metholdogy section is vague.
4. There are only names of features in the “feature extraction and selection” section. The details of features and extraction process should be presented clearly in the manuscript.
5. In line 296, the authors discussed eight classifiers and named them in result sections. However, algorithm, hyper parameter, and grid search details are not presented anywhere in the manuscript.
6. It would have been better to perform a comparative analysis among eight classifiers and previously established approaches, which the authors discussed in “Survey Methodology”.
7. The results should be discussed in detail along with solid proofs (e.g., AUC-ROC plots, Important feature plots.)
8. The evaluation metrics details are missing.
9. The figure quality is poor; they need to be improved.
10. Line 320: “The ultimate goal of this research project is to develop….”. change this to “The ultimate goal of this research project was to develop….”
11. The conclusions section is detailed, whereas the Discussion is too short. Try Discussing more and shorten the conclusions.

Experimental design

The work is timely, well thought and executed;

Validity of the findings

Impactful and novel

Reviewer 2 ·

Basic reporting

1. The manuscript needs to be professionally written. There are several grammar and technical errors that need to be corrected. It would be advisable that authors request someone with excellent English language to read and edit the manuscript before submitting it.
2. References are adequate but there need to be better cited in the main text. Several methods references are poorly referenced in the manuscript.
3. While the manuscript has standard sections, the sections are not adequately described. The figures are not adequate and are of poor quality. I think there is no requirement for data sharing, as all data is previously published by the authors.
4. The hypothesis or the objective of this study is not clearly defined.
5. Results are very poorly described as seen in my detailed comments below.

Experimental design

The manuscript is poorly written and the methods and experimental design are poorly described. Therefore, it is difficult to comment on the validity of the results obtained.

Validity of the findings

Since the methods are poorly defined, it is difficult to understand what authors are trying to achieve and therefore not possible to evaluate their results. Please see my detailed comments below:
* Line 62-66: Not a strong argument for imaging platforms. Before this reason give what the current challenges to higher wheat production and which of these can be solved by high throughput phenotyping technologies.
* Line 71: Data about Southern India may not be relevant for wheat as rice is the main crop and also no reference for the statement.
* Line 80-90: This para is about plant physiology and photosynthesis. It is not clear what authors what to state in this para with regards to the topic of their research in this paper. Also, Figure 1 is not required and not relevant.
* Line 92: Which 3 approaches?
* Line 113-120: This para is irrelevant to the topic.
* Line 122-134: The literature review needs to be properly written rather than just a list of previous work. It should mention how previous researchers have tried to solve this problem. Since it s a comparison study, a comparative analysis of each technique will be good.
* Line 136: "The authors...." the manner of using references in the main text is not correct. In this case, it should be "Hassan et al. [28]". This has to be corrected throughout the manuscript.
* Line 166-180: Authors describe several techniques while trying to compare high throughput techniques, but in this section, they are describing techniques which are not high throughput. Hence, confused about how it is relevant to the current study.
* Line 201: What is Lagunas? If it is a spelling error, there are several grammatical mistakes throughout the manuscript.
* Line 228: The image dataset used by the authors is too small - only 1440 images. It is not clear how such a small dataset was used for so many ML algorithms that usually require larger datasets.
* Line 253-257: This para is completely confusing.
* Line 262-263: Not clear how these features were identified?
* Line 267: Again not clear how 23 features were identified?
* Line 274-278: Authors refer to as et a 24 images. Are these samples or the complete dataset? If they are samples, where is the complete dataset? Also, it is not clear what conditions these plants were grown, how many plants were imaged, etc.?
* Line 285-289: Since the authors have described their methods properly, it is difficult to understand any of their analyses. No description details are given for any statistical or analytic methods.
* Line 291: How is the combination determined to be the best out of so many features? It again could be because of a lack of detailed methods missing the manuscript.
* Line 293: Where is this correlation matrix?
* The whole results section is very poorly described and it is not possible to understand any of their statements. On what basis do they find that SVM, ETC and KNN are the best methods? Again because of poor details, it is difficult to assess these results.
* Discussion and conclusion sections are too short and do not contextualise the results, obviously!
*Line 353-359: looks like results description rather than conclusions.
* Figs 1 and 2 are not useful at all.
* Fig 3 describes results but there is no discussion of images here in the result section. There is no description of ground truth data or baseline.
* Fig 4: Unfortunately this figure has more details than the whole methods section.
* Fig 5: Completely unreadable figure
* Fig 6: What is this result? If yes, where is it described or referenced in the main text?
* Table 1: No clue what this table describes?
* Table 2: Again incomplete table in terms of description and what parameters were used for each algorithm?

Additional comments

None

Reviewer 3 ·

Basic reporting

Overall the study is built upon a good hypothesis and important scientific questions that AI can answer for agriculture, however, there are many questions and comments:-

Line 43: The abstract can't start with a sentence on "this research effort" without even mentioning what is the study about.
The manuscript requires grammar corrections.
Expand abbreviations used in the abstract.
Give references for the first three statements in the Introduction section, i.e. "Wheat is one of the world's ..... after China".
While referring to authors in the main text with a reference, write the names of the first author. For example, lines 136, 139, 150, etc.
Few references are incomplete. For example, names of journals/books are missing in reference numbers 3, 4862, 63, etc.
Remove random use of capitalized first letters in words throughout the manuscript.
The review figures are ok.

Experimental design

Research questions are well defined and under the scope of the journal.
In the methods section:-
Give justification for using a small sample of images and low-resolution images for the study.
Include justification for preprocessing of the images and why only a certain kind of preprocessing is used.
Describe the size of the final training dataset size. Instead you mention towards the end that 2880 instances of images for both control and drought.
What was the basis for the selection of 8 ML algorithms?
Which software was used for ML model generation?
Line 232: the data deposition is not part of the current study. please correct this.
Line 253: Elaborate on how the features are suitable for "this purpose".
How are the features correlated with the fluorescent measurements in the study?

Validity of the findings

Discussion:
Line 331: What is data "argumentation"? Do you mean augmentation? Was it employed in the current study? If yes, where are the details?
There is no external validation of the SVM model. How do you justify this?
Data argumentation in line 357.
Line 358: why did not you use data augmentation in this study?
What are the main limitations of the study with respect to using the method on different geographical locations?

Additional comments

Line 366: Data is not available with the submission, it is available from a public database.
The final training dataset (after pre-processing) must be made available for evaluation of the models and reproducing the study.
Line 369: Codes are not available with the manuscript. If available from any public repository it must be mentioned and links given.

---

## Round 0.2 · Minor Revisions

The reviewer comments should be addressed; seasonal conditions for drought stress study and related literature should be surveyed

Reviewer 1 ·

Basic reporting

The revised version is much improved. All my concerns are well addressed

Experimental design

The revised version is much improved. All my concerns are well addressed

Validity of the findings

The revised version is much improved. All my concerns are well addressed

Additional comments

The revised version is much improved. All my concerns are well addressed

Reviewer 4 ·

Basic reporting

English in abstract and conclusion section has to be reviewed to clearly specify the research work findings. Background about cause of drought and how one can predict need to be added. Beside this some of the existing crop models applicable need to be discussed.

The figures are relevant but name of algorithms with its short form should be consistent throughout the paper including figure and tables.

What other kind of data are available and used in developing and validating model for drought on wheat to be referred or specified.

Experimental design

Some of the crucial factors for differentiating stressed and non-stressed crops need to be specified. The problem taken under consideration is well defined in the research paper with support of experimental results but some of the characteristics that are most helpful in a model for detecting the effects of drought need to be specified.

Validity of the findings

Technological and sociological impacts of the study need to be emphasized in the discussion and conclusion. Some statistics regarding the normal seasonal drought stress that occurs in areas where wheat is grown need to be provided.

Additional comments

The study work that has been done has taken drought conditions into consideration. As a result, it is strongly suggested that the following research topics be discussed in the introduction and the literature review:

1) What are the causes of droughts, and how can one predict that when it will occur?
2) Which of the already-existing crop models are applicable to the identification of drought stress in wheat?
3) What kinds of data are available for use in developing and validating models of the effects of drought on wheat?
4) Which characteristics are the most helpful in a model for detecting the effects of drought?
5) In order to differentiate stressed crops from non-stressed crops, what do you consider to be the most crucial factors?
6) Could you provide some statistics regarding the normal seasonal drought stress that occurs in areas where wheat is grown?

Second, both the technological and sociological impacts of the study need to be emphasized in the discussion and conclusion portions of the paper.


Third, there must be consistency in the name and abbreviation used for ML Algorithms throughout the paper e.g either NB - Naïve Bayes or Gaussian Naïve Bayes (GNB), Similarly, DT – Decision Tree or Decision tree (CART), GB or GBC, KNN – K – Neighbours Classifier or K-Nearest Neighbours etc

Annotated reviews are not available for download in order to protect the identity of reviewers who chose to remain anonymous.

---

## Round 0.3 · accepted · Accept

I have read the paper and authors have addressed all the comments raised by the reviewer - I recommend acceptance.